# Investigating In Vitro and Ex Vivo Properties of Artemether/Lumefantrine Double-Fixed Dose Combination Lipid Matrix Tablets Prepared by Hot Fusion

**DOI:** 10.3390/pharmaceutics13070922

**Published:** 2021-06-22

**Authors:** Christi A. Wilkins, Lissinda H. du Plessis, Joe M. Viljoen

**Affiliations:** Centre of Excellence for Pharmaceutical Sciences (PharmacenTM), Building G16, North-West University, 11 Hoffman Street, Potchefstroom 2520, South Africa; christiwilkins09@gmail.com (C.A.W.); Lissinda.DuPlessis@nwu.ac.za (L.H.d.P.)

**Keywords:** artemether, lumefantrine, lipid-based formulations, solid lipid dispersion, hot fusion, double-fixed dose combination, biorelevant media, dissolution

## Abstract

Highly lipophilic antimalarial drugs, artemether and lumefantrine, whilst an effective fixed-dose combination treatment to lower the malarial disease burden, are therapeutically hindered by low aqueous solubility and varied bioavailability. This work investigates the plausibility of directly compressed lipid matrix tablets, their role as lipid-based formulations and their future standing as drug delivery systems. Lipid matrix tablets were manufactured from solid lipid dispersions in various lipid:drug ratios employing hot fusion—the melt mixing of highly lipophilic drugs with polymer(s). Sequential biorelevant dissolution media, multiple mathematical models and ex vivo analysis utilizing porcine tissue samples were employed to assess drug release kinetics and more accurately predict in vitro performance. Directly compressed stearic acid tablets in a 0.5:1 lipid:drug ratio were deemed optimal within investigated parameters. Biorelevant media was of immense value for artemether release analysis, with formulation SA0.5C1 (Stearic Acid:double fixed dose in a 0.5:1 ratio (i.e., Stearic acid 70 mg + Lumefantrine 120 mg + Artemether 20 mg); CombiLac^®^ as filler (q.s.); and 1% w/w magnesium stearate) yielding a higher percentage of artemether release (97.21%) than the commercially available product, Coartem^®^ (86.12%). However, dissolution media lacked the specificity to detect lumefantrine. Nonetheless, stearic acid lipid:drug ratios governed drug release mechanisms. This work demonstrates the successful utilization of lipids as pharmaceutical excipients, particularly in the formulation of lipid matrix tablets to augment the dissolution of highly lipophilic drugs, and could thus potentially improve current malarial treatment regimens.

## 1. Introduction

The World Health Organization has called for the use of artemisinin-based combination therapy in the fight against malaria, which accounted for an estimated 405,000 deaths globally in 2018 [1]. The Biopharmaceutics Classification System (BSC) has categorized the highly lipophilic antimalarial drugs, artemether and lumefantrine, in Class II (drugs having high permeability but low solubility). These drugs are furthermore employed as a double-fixed dose combination in a 6:1 ratio (lumefantrine:artemether) for first-line antimalarial therapy in commercial products such as Coartem^®^ [2,3,4]. Overall, the therapeutic potential of highly lipophilic drugs is considerably hindered due to their low and inconsistent bioavailability arising from poor aqueous solubility. Thus, the primary challenge remains to design a dosage form capable of enhancing the solubility of these active pharmaceutical ingredients.

Low aqueous solubility can chiefly be ascribed to high intermolecular forces contained within a crystal lattice, high lipophilicity, or a combination of these elements [5]. One method implemented to augment the dissolution rate and subsequent bioavailability of such compounds, is hot melt extrusion. Hot melt extrusion is the process of applying heat and pressure to melt a material—for example, a polymer—which is forced through an orifice in a continuous process. It is proposed to enhance dissolution by increasing the surface area and saturation solubility [5]. Hot fusion involves heating a polymer in the absence of pressure application to the molten mass. These methods of manufacture are well suited for the preparation of lipid-based formulations (such as lipid matrix tablets) which operate on the premise that ingested exogenous lipids containing active pharmaceutical ingredients will be processed by bile salts and phospholipids into mixed micelles and colloidal species [6,7,8,9,10]. Thus, lipid-based formulations rely on natural physiological responses to the presence of lipids in the gastrointestinal tract (GIT) to provide a microenvironment into which lipophilic drugs may partition and subsequently be shuttled to the enterocytes of the intestinal wall for absorption [11].

Two contributing factors responsible for the intermixing of drug–polymer particles at a molecular level are the glass transition temperature of both the polymer and active pharmaceutical ingredient, as well as the melting point of said active pharmaceutical ingredient. The higher apparent solubility and increased dissolution rate for amorphous materials are extensively documented [12,13,14,15,16]. Their increased dissolution rates are linked to the lower thermodynamic barrier to dissolution and the formation of a glassy solution wherein the active pharmaceutical ingredient is molecularly dispersed within the polymer. The enhancement in solubility is as a result of the disordered structure of the amorphous solid. Moreover, crystalline material requires the disruption of the crystal lattice as a prerequisite for dissolution, whereas an amorphous system has short-range intermolecular interactions that require no lattice energy to be overcome. Therefore, when employing hot fusion as a method of manufacture, it is important to consider the melting point, glass transition temperature, and polymorphic behavior of the active pharmaceutical ingredients, as well as the dispersion medium selected [17].

A notable application for dissolution testing is to predict the in vivo performance of a solid oral dosage form. By identifying the limiting factor (solubility, dissolution rate or permeability), as described by the BCS, it may be possible to design a dissolution study tailored to the needs of the formulated dosage form and respective drug components [18,19]. Many dissolution studies are not representative of all the physiological aspects in the GIT, and the extent of inadequate aqueous solubility for BCS Class II and IV drugs is often overstated; this solubility is determined in compendial dissolution media consisting primarily of purified water and buffers. The degree of aqueous solubility is therefore misleading, as these tests do not account for the significantly higher solubilizing capacity of the human intestine as a result of the presence of bile salts [19]. Biorelevant media can more accurately predict in vivo performance, as they consider both the physiological conditions of the GIT as well as the properties of the drug and dosage form. Predictions of a dosage form’s intraluminal performance requires adequate simulation of the stomach and proximal part of the small intestine, which necessitates the incorporation of gastrointestinal (GI) fluid properties, such as composition, volume, pH, gastric emptying (particularly important for non-disintegrating systems), GI enzymes and the presence or absence of food [20]. Sunesen et al. [21] reported that compendial dissolution media often fail to yield in vitro–in vivo correlations for Class II components due to the absence of relevant physiological parameters. Therefore, concluding that, when optimizing a dissolution media’s composition, it is crucial that the physiological relevance is of prime consideration, i.e., biorelevant media are utilized. Additionally, Wang et al. [22] proposed the use of biorelevant media with the incorporation of lipolytic products for lipid-based formulations, predominantly dosage forms containing BSC Class II or IV compounds, due to drug solubilization and formulation properties that have a substantial effect on the in vitro–in vivo correlation.

There are several matrix-forming materials available, depending on the properties of the drugs to be incorporated into these drug delivery systems. Thus, the physical characterization of active pharmaceutical ingredients and the selection of appropriate matrix-forming materials to be subjected to hot fusion or hot melt extrusion are vital. The lipid bases, stearic acid (C_17_H_35_CO_2_H), glycerol monostearate (C_21_H_42_O_4_) and cetyl alcohol (CH_3_(CH_2_)_15_OH) were chosen for this study, as they are all long-chain lipid bases with differing melting points and matrix-forming properties. Utilizing long-chain fatty acids enhances lymphatic uptake of lipid-based drug delivery systems, which can assist in avoiding the hepatic first-pass metabolism of an active pharmaceutical ingredient [23,24]. Findings relating to the physical characterization of these active pharmaceutical ingredients and the resulting effects of preparing solid lipid dispersions by means of hot fusion have been reported by the authors in previous work. Powder flow characterization, as well as physical observations utilizing X-ray powder diffraction studies, thermogravimetric analysis and differential scanning calorimetry, were conducted and discussed. It was concluded that hot fusion of the various lipids served only as coating of the active ingredients and did not obscure their crystallinity. Additionally, the produced solid lipid dispersions demonstrated significantly improved powder flow properties in relation to the individual active ingredients, based on an increase in overall particle size, as well as a more spherical shape. For more detail, the interested reader is referred to work by Wilkins et al. [10].

Overall, the present study built on this previous work [10] and sought to design and determine the feasibility of directly compressed lipid matrix tablets containing a double- fixed dose combination of artemether and lumefantrine. Objectives included characterizing the impact of hot fusion when manufacturing lipid matrix tablets and evaluating the biopharmaceutical behavior of the matrix system. The focus was on the relevance of biorelevant media and the resulting drug release mechanisms where, afterwards, ex vivo experiments were conducted.

## 2. Materials and Methods

### 2.1. Materials

Artemether was purchased from DB FINE CHEMICALS (Pty) Ltd. (Johannesburg, South Africa); lumefantrine was obtained from Cipla (Pty) Ltd. (Cape Town, South Africa). Stearic acid, glycerol monostearate, acetonitrile and octane-sulphonic acid were acquired from Associated Chemical Enterprises (Pty) Ltd. (Johannesburg, South Africa). Cetyl alcohol and methanol were procured from Merck (Darmstadt, Germany), and Coartem^®^ was purchased from Novartis (Pty) Ltd. (Johannesburg, South Africa). CombiLac^®^ and MicroceLac^®^100 were obtained from Meggle (Meggle Group, Wasserburg, BG Excipients & Technology, Wasserburg, Germany). All other chemicals employed were of analytical grade.

### 2.2. Preparation of Solid Lipid Dispersions Utilizing Hot Fusion

Fixed-dose solid lipid dispersions comprising 120 mg lumefantrine and 20 mg artemether were prepared by means of the hot fusion method [10]. The individual lipid bases (stearic acid, glycerol monostearate or cetyl alcohol) were melted by means of continual stirring in a porcelain dish in a water bath maintained at 75 °C (±0.5 °C). Active pharmaceutical ingredients, in their fixed ratios, were added to the melted lipid in a predetermined lipid:drug ratio (Table 1), and stirred until a homogenous mixture was obtained [23,24]. The molten mass was allowed to cool and solidify at room temperature (±25 °C) where, afterwards, the hardened mass was manually crushed with a pestle and mortar and screened through a 595 μm sieve. The resulting powdered solid lipid dispersions were individually stored at 25 °C (±0.5 °C) in glass containers, sealed with Parafilm^®^, until utilized for tablet manufacture (i.e., within 48 h).

### 2.3. Full Factorial Experimental Design

The complete experimental design of the independent and dependent variable levels applied for this study are shown in Table 1, where, X_1_ signifies the lipid base (independent variable) and X_2_ denotes the lipid:drug ratio (dependent variable) [10]. The lipid bases were assigned the following notations: stearic acid (−1), glycerol monostearate (0) and cetyl alcohol (1). Each of these variables were evaluated on three levels regarding the lipid:drug ratio, i.e.: 0.5:1 (low level, −1), 0.75:1 (intermediary level, 0) and 1:1 (high level, 1). Dependent variables further included filler composition X_3_: CombiLac^®^ (0) and MicroceLac^®^ 100 (1), as well as lubricant concentration X_4_: 1% (0) or 1.25% (1).

Abbreviation codes were allocated to the combinations of factors measured to aid in-text referencing and are notated in parenthesis as follows: stearic acid (SA), glycerol monostearate (GM) and cetyl alcohol (CA). The aforementioned two-letter designation signifying the type of lipid employed as the base of the solid lipid dispersions is followed by a numerical value indicating the lipid:drug ratio as either 0.5-, 0.75- or 1-part lipid in relation to 1 part fixed-dose combination. This numerical value is subsequently followed by the first letter of the filler type (CombiLac^®^-C; MicroceLac^®^100-M), followed lastly by the concentration lubricant (1 or 1.25). Consequently, the following acronym, SA0.5M1, denotes stearic acid (SA) in a 0.5:1 ratio with the fixed artemether and lumefantrine dose combination, MicroceLac^®^100 (M) as filler, and 1% magnesium stearate (MgSt).

### 2.4. Preparation of Directly Compressed Tablet Formulations

Solid lipid dispersions were prepared by weighing the respective active pharmaceutical ingredients (APIs) in the World Health Organization (WHO) approved ratio of 6:1 containing 120 mg lumefantrine and 20 mg artemether. The fixed dose combination was subsequently combined with a selected lipid in a corresponding lipid:drug ratio of either: 0.5:1; 0.75:1 or 1:1. Three lipid bases namely, glycerol monostearate, stearic acid and cetyl alcohol, were investigated utilizing the previously mentioned full factorial design of experiments. All of the solid lipid dispersions were mixed with the relevant excipient combinations (Table 1) for 7 min in sealed glass containers using a Turbula^®^ mixer (T2C, W.A., Bachofen AG Maschinenfabrik, Bastle, Switzerland), as per the full factorial experimental design. Powder mixtures were tableted by means of direct compression utilizing a Korsch^®^ XP1 single station tablet press (Korsch^®^, Berlin, Germany), 10 mm concave faced tableting punches (final weight of approximately 500 mg) and a stroke rate of 20 strokes/min.

### 2.5. Pharmacotechnical Characterization

To assess the mass variation, the individual weight of 20 randomly selected tablets for each formulation was measured on an analytical balance (Precisa^®^, Zurich, Switzerland). Friability was performed employing a friabilator (ERWEKA^®^ GmbH, Heusenstamm, Germany) according to the British Pharmacopoeia (BP) specifications [25] that states that tablets should be arbitrarily selected from a formulation to a weight as near as possible to 6.5 g. This was approximately 10 tablets for each formulation. Disintegration testing was conducted on all formulations utilizing an Erweka^®^ disintegration apparatus (model D-63150, Heusenstamm, Germany) where 6 randomly selected tablets from each formulation were tested. Following, the crushing strength, diameter, thickness and tensile strength of 10 indiscriminately selected tablets were established using a Pharma Test^®^ tablet test unit (model PTB-311, Schlieren, Switzerland). Samples (*n* = 10) from each formulation were compressed at a rate of 0.1 cm·min^−1^. All pharmacotechnical characterization was performed in triplicate according to the BP specifications [25] for tablets with an average weight exceeding 250 mg. Acceptance criteria for all experiments were in accordance with those specified within the BP [25]. However, a minimum tablet hardness of 60 N [26] was deemed acceptable. Additionally, tablets that remained intact after a 15 min experimental exposure period during disintegration analyses were considered suitable, as these tablets showed signs of potential modified release, which was required for this study. Standard deviation (SD) and percentage relative standard deviation (%RSD) were furthermore calculated where applicable. Each formulation was tested in triplicate per analysis.

### 2.6. Assay

Twenty tablets per formulation were crushed and 500 mg was weighed from the powdered mass. Each powder sample was dissolved in 100 mL MeOH and 1 mL orthophosphoric acid, and continuously stirred for 15 min, after which it was ultrasonicated in a Labotec Ecobath^®^103 (Labotec, South Africa) for 20 min. The solution was filtered through a 0.45 µm membrane filter and the resulting filtrate was diluted to 200 mL. This solution was subsequently analyzed by means of high performance liquid chromatography (HPLC), as validated by Costa et al. [27], employing an Agilent 1100 HPLC system (Agilent Technologies, Santa Clara, CA, USA) equipped with a Luna C18-2 column, 150 × 4.6 mm, 5 µm column (Phenomenex, Torrance, CA, USA). The mobile phase consisted of acetonitrile (85% *v/v*) and octane-sulphonic acid (15% *v/v*) at pH 3.5, with a flow rate of 1 mL/min and detection wavelengths of 210 nm and 303 nm for artemether and lumefantrine, respectively. Each formulation was tested in triplicate (i.e., *n* = 3).

### 2.7. Dissolution Behavior

The effect of the transference from gastric to intestinal pH on drug release was determined in sequential dissolution conditions with biorelevant media present. An initial stirring rate of 100 rpm was set in a Distek^®^ dissolution system (model 2500, Distek^®^ Inc., North Brunswick, NJ, USA), connected to a Distek^®^ Evolution 4300 auto sampler (model 4301920) and a Distek^®^ syringe pump (SP02716), as per the BP basket method [25]. The six baskets (*n* = 6) were introduced into the medium at time zero (*t* = 0). The sequential protocol was as follows: 600 mL starting solution, pH 1.2 for the first 2 h, followed by the addition of 300 mL 0.2 M trisodium phosphate dodecahydrate buffer (Na_3_PO_4_), pH 6.8 for 3 h and, finally, the addition of 3 mM bile salts and 0.5 mM phospholipid at a time interval of 300 min for 7 h at a pH 7.4 [28,29]. At predetermined time intervals (2, 5, 10, 20, 30, 60, 90, 120, 150, 180, 240, 300, 390, 480, 600 and 720 min; plus, an infinity sample at 150 rpm for 30 min), withdrawn samples were analyzed by means of HPLC.

### 2.8. Analysis of Drug Release Mechanism

Artemether and lumefantrine release kinetics were evaluated using mathematical models classically employed to assess modified drug release profiles by means of DDSolver software (a freely available menu-driven add-in programme for Microsoft^®^ Excel™ 2016 for Windows™; Microsoft^®^ Corporation, Seattle, Washington, USA) [30]. Three selection criteria determined the goodness of fit of a model, as well as the mechanistic plausibility of the model, namely the adjusted coefficient of determination (Rsqr_adj), the Akaike Information Criterion (AIC) and Model Selection Criterion (MSC) [30,31]. Additionally, the release exponent (n) was correlated to identify the mechanism of drug release, i.e., Fickian diffusion, non-Fickian diffusion, Case II transport or Super Case II transport. For an in-depth understanding of the above selection criteria, the interested reader is referred to work by Zhang et al. [30].

### 2.9. In Vitro Permeability Studies

Ethical approval for the procurement of excised porcine jejunum segments from a local abattoir was obtained from the Health Research Ethics Committee, North-West University (NWU), South Africa (approval number: NWU-00369-16-A1, approval date: 07.12.2016). The serosa layer of excised porcine jejunum segments was removed by means of blunt dissection. Excised tissue was rinsed and kept moist with cold Krebs-Ringer bicarbonate (KRB) buffer for the duration of experimentation. Intestinal segments were incised along the mesenteric border, and smaller segments (8 cm × 2 cm -apical side upwards) were cut and mounted onto Sweetana-Grass diffusion chamber inserts (Easy Mount Diffusion Chamber, Physiological Instruments, San Diego, CA, USA). The basolateral half-cells protocol was as follows: heating block containing 7 mL KRB buffer per chamber, heated and maintained at 37 °C, parallel gas flow (5% CO_2_, 95% O_2_; flow rate: 15–20 mL/min), allowed 15 min to equilibrate. Thereafter, KRB buffer was removed from the apical chamber via aspiration with a vacuum system (Vacusafe^®^, Hudson, NY, USA) and replaced with 7 mL pre-heated KRB buffer containing the dosage form to be analyzed, as well as 3 mM bile salts and 0.5 mM phospholipids to simulate the systemic environment.

Samples (1 mL) from each receiver compartment were withdrawn at 20 min intervals (duration 2 h) for triplicate analysis (*n* = 3) in the apical to basolateral (AP-BL) direction. These samples were stored at 4 °C pending analysis by means of HPLC and an equivalent volume of heated media was replacement immediately. Trans-Epithelial Electrical Resistance (TEER) readings were taken directly before the addition of the lipid matrix tablets to the apical chamber, and subsequently every 20 min per chamber to continually evaluate tissue integrity using a Dual Channel Epithelial Voltage Clamp (Warner Instruments, Hamden, CT, USA) for the duration of the permeation study.

Ex vivo analysis involved removal of the tissue sample from the Sweetana-Grass diffusion chamber post in vitro testing. It was then rinsed with deionized water, followed by placement into a 15 mL glass vial containing 5 mL of 100% HPLC grade MeOH. Vials were mixed using a Vortex^®^ mixer, placed in an ultrasonic bath for 10 min to cause lysis of the tissue, and subsequent centrifugation for 5 min at 3000 rpms to separate the supernatant. Subsequently, 500 µL samples were withdrawn from the supernatant and filtered (45 µm) for HPLC analysis, which was performed in triplicate (*n* = 3). The percentage of active pharmaceutical ingredients present in the tissue post 120 min was determined and presented as percentage retention.

### 2.10. Statistical Data Analysis

The mean dissolution time (MDT), as well as fit factors [32], were calculated. Following this, the dissolution profiles of the test formulations and the control (Coartem^®^) were compared and discussed according to their MDT and fit factor values. Furthermore, the fit factors concerning interrelating formulations are reported in the Appendix A. The MDT indicates the average time it will take for the entire drug dose to be released from the dosage form into solution (Equation (1)).
(1)$$\mathrm{MDT}=\frac{\sum_{j=1}^{n}t_{\mathrm{mid}\;}\Delta x_{d}}{\sum_{j=1}^{n}\Delta x_{d}}$$
where j is the sample number; n is the total number of samples; t_mid_ is midpoint time between j and j − 1; and Δx_d_ is the additional mass of drug dissolved between j and j − 1.

Fit factor f_1_ is the difference factor (Equation (2)) and was utilized to determine the percentage error between the two curves. Indistinguishable curves are represented by a value of 0. Fit factor f_2_ is the similarity factor (Equation (3)) between the two curves and is a logarithmic transformation of the sum of squares error. A value of ≥50 indicates that both the test and control formulations are similar, with a value of 100 showing that the two samples are identical [33].
(2)$$f_{1}=\frac{\sum_{j=1}^{n}\left|R_{J}-T_{J}\right|}{\sum_{j=1}^{n}\left(R_{J}-T_{J}\right)/2}\times 100$$
(3)$$f_{2}=50\times \log\left\{{\left[1+\left(\frac{1}{n}\right)\sum_{j=1}^{n}{\left|R_{J}-T_{J}\right|}^{2}\right]}^{-0.5}\times 100\right\}.$$
where R_J_ is the reference assay at time point t. T_J_ is the test assay at time point t and n is the number of pull points.

Additionally, the percentage active pharmaceutical ingredients transported across the excised intestinal tissue was plotted as a function of time, and the apparent permeability coefficient values were calculated using Equation (4) [34,35].
(4)$$P_{\mathrm{app}}=\frac{\mathrm{dQ}}{\mathrm{dt}}\left(\frac{1}{A.C_{0}60}\right)$$
where P_app_ denotes the apparent permeability coefficient (cm·s^−1^) and dQ/dt (µg/s) represents the increase in the amount of drug in the receiver chamber within a given time period (i.e., the permeation rate as µg·s^−1^), which is equivalent to the slope of the plot of drug concentration transported versus time. A (cm^2^) signifies the effective surface area of the excised porcine intestinal tissue section between the apical and basolateral chambers, and C_0_ (µg·mL^−1^) is the initial concentration of the specific compound present in the apical chamber.

Statistical analysis was conducted using Statistica software (ver.12; TIBCO Software Inc., New York, NY, USA). One-way analysis of variance (ANOVA) was performed, where *p* values of ≤0.05 were considered statistically significant.

## 3. Results and Discussions

### 3.1. Analysis of Physical Tablet Properties

The pharmacotechnical tablet properties of the various lipid matrix tablet formulations were researched (Appendix A) and compared employing a full factorial design to assess each factor systematically and impartially. A summary table comparing all the factors evaluated at each level, and their corresponding responses per test, may be viewed in Table 2.

Analysis indicated that all formulations, regardless the lipid base employed, in combination with either MicroceLac^®^100 or CombiLac^®^, produced lipid matrix tablets that overall adhered to the specified BP [25] criteria. During preformulation studies, RetaLac^®^ and Pharmacel^®^101 were additionally analyzed as possible fillers for inclusion in lipid matrix tablet formulations. However, tablets comprising these fillers lacked the required mechanical strength, and the pharmacotechnical results obtained furthermore skewed the performance characterization data of the other dependent factors. We therefore excluded reporting these results; however, relevant data are provided in the Appendix A.

A prerequisite for this study was modified drug release, which was set to be achieved by the hydrophobic nature of the lipid bases employed. Thus, formulations were required to remain intact post 15 min disintegration testing (unlike for conventional immediate release tablets). Generally, lipid matrix tablet formulations complied with this objective due to the inclusion of a lipid base and demonstrated no other factor-dependent disintegration.

Stearic acid comprising formulations generally depicted the smallest average mass variation (%RSD = 1.027), an average tensile strength of 2.031 N·mm^−2^ and satisfactorily low percentage friability (<0.8%), whilst glycerol monostearate formulations evaluated at filler level displayed the lowest average tensile strength values (Table 2). Stearic acid is often employed as a thickening and hardening agent in the cosmetic industry, which may account for the increased tablet hardness presented as superior tensile strength in comparison to the other lipid bases. Considering all pharmacotechnical characteristics, the following general rank order could be allocated to the type of lipid bases investigated: SA > GM > CA.

Comparisons of the various lipid:drug ratios and their corresponding average friability and tensile strength values yielded a distinct trend that an increase in lipid concentration caused a subsequent decrease in tensile strength and a proportional increase in percentage friability (Figure 1). This synchronous relationship between tensile strength and friability can be attributed to tablets with a higher tensile strength, i.e., with stronger interparticle forces and tighter compaction, rendering less friable tablets.

Moreover, the 0.5:1 lipid:drug ratio produced tablets with the highest average tensile strength (2.084 N·mm^−2^) and an adequately low friability (0.459%); however, overall, it exhibited the highest average mass variation (%RSD = 2.088%). The lower lipid content dictated higher filler content to achieve the target tablet weight, which could account for the higher %RSD-values due to the compaction properties and smaller particle size distribution of the fillers versus the solid lipid dispersions [10]. Although the 0.5:1 ratio percentage friability was in accordance with the BP standards [25], the higher tensile strength caused the tablets to be notably brittle, accounting for the higher percentage friability compared to that of the 0.75:1 ratio. The 1:1 ratio demonstrated the lowest tensile strength (1.735 N·mm^−2^) and the highest percentage friability, implying that the reduction in tablet hardness caused these formulations to be more friable. Friability is a measure of a tablet’s resistance to abrasion during packaging, transportation and handling, and is strongly correlated with compression force. Highly friable tablets often relate to a low compression force, resulting in weak bonding between adjacent particles. Particles in closer proximity to one another form stronger bonds, and therefore have increased mechanical strength [36]. Pertaining to the pharmacotechnical characteristics obtained for the investigated lipid:drug ratios, the following ranking was assigned: 0.75:1 ≥ 0.5:1 > 1:1.

Magnesium stearate was included to aid the compaction process due to its ability to reduce wall friction during tablet ejection [37], improve powder flow [38] and reduce the risk of pharmaceutical formulations adhering to exposed metal surfaces during tableting [39,40]. Formulations containing 0% magnesium stearate adhered to the tablet punches during tableting, resulting in a non-uniform appearance and tablet weight (preformulation research). Furthermore, there was no substantial difference in tablet characteristics, regardless of the inclusion of magnesium stearate, at two different levels (1% and 1.25%). All formulations containing either 1% or 1.25% magnesium stearate produced acceptable average %RSD-values of 1.411% and 1.694%, respectively, for mass variation, and yielded no noticeable differences in the internal or external morphology of manufactured tablets (Appendix A). However, an increase in magnesium stearate concentration did result in an increase in %RSD-values and an even more pronounced increase in the percentage of friability. The occurrence of an increase in magnesium stearate concentrations, rendering decreased tablet hardness and increased friability, is well documented [37,41,42,43,44]. Additionally, an equal average tensile strength of 1.914 N·mm^−2^ was achieved when evaluating magnesium stearate concentration as a factor (Table 2), which suggests that the difference in magnesium stearate concentration was insufficient to produce a significant variance in response when assessed at filler and lipid:drug ratio level. For these reasons, 1% magnesium stearate was identified as the lowest functioning lubricant concentration necessary to produce acceptable lipid matrix tablets in this study.

### 3.2. Assay

Artemether and lumefantrine content was analyzed for each formulation (Appendix A). To be able to ascertain whether noticeable differences were present, a summary table of average artemether and lumefantrine content per factor was constructed (Table 3). 

No conspicuous differences were detected between the various lipid bases for artemether content. However, when stearic acid was employed, lumefantrine content was deliberated as the highest when comparing lipid bases. MicroceLac^®^100 formulations overall depicted slightly higher active pharmaceutical ingredient content for both active ingredients. The 0.5:1 lipid:drug ratio similarly displayed a higher drug content for both active pharmaceutical ingredients, whereas the 1:1 ratio exhibited the lowest active pharmaceutical ingredient content of the three ratios investigated. Thus, as lipid concentration increased, the overall active pharmaceutical ingredient content appeared to decrease.

The overall trend observed revealed that the percentage of artemether content obtained was typically lower than the theoretical content whilst, in comparison, a noticeably higher percentage of lumefantrine content was demonstrated. The variance in target content may have resulted from loss of active pharmaceutical ingredients due to dust formation during post-hot fusion processing of solid lipid dispersions and lipid matrix tablet manufacture.

### 3.3. Dissolution Behavior

Stearic acid-containing lipid matrix tablet formulations displayed a lipid:drug ratio-dependent artemether release trend (Figure 2a). Moreover, the 0.5:1 ratio, regardless of the filler, fitted the Korsmeyer–Peppas with T_lag_ model, which is indicative of Fickian diffusion (*n*-value < 0.45). The 0.75:1 ratio fitted the Peppas–Sahlin1 with T_lag_ model, consistent with non-Fickian diffusion that entails a diffusion rate higher than or equal to the rate of polymer relaxation. Finally, the 1:1 ratio conformed to the Peppas–Sahlin2, with the T_lag_ model demonstrating the drug transport mechanism was controlled by both Fickian diffusion and Case II relaxation.

SA0.5C1 was identified as the optimal formulation in this study regarding artemether release (97.21% of theoretical content), and was compared to the artemether dissolution results achieved from Coartem^®^ tablets (86.12% artemether pharmaceutically available). Statistically relevant differences between the lipid:drug ratios (f_1_- and f_2_-values, Table 4) were present between SA1C1 and SA0.5C1, SA1C1 and SA0.75C1 as well as SA1M1 and SA0.75M1. In addition, comparisons between formulations of the same lipid:drug ratios but different filler combinations for stearic acid also revealed statistically significant differences, signifying that the filler type, together with the lipid:drug ratio, were instrumental in drug release (Appendix A).

Glycerol monostearate containing lipid matrix tablet formulations displayed the highest variance in lag time, as well as percentage of artemether released between the respective formulations when compared to stearic acid lipid bases (Figure 2b). GM0.75M1 specifically displayed release characteristics dependent upon the presence of the biorelevant media. GM1M1 also stressed the influence of biorelevant media, as there are two distinct spikes in drug release: first, when the pH was increased (120 min), and second, with the addition of biorelevant media (300 min). All glycerol monostearate containing formulations displayed statistically relevant differences (*p* < 0.05) when compared (Table 4; Appendix A). They additionally portrayed the lowest average mean dissolution times of the lipid bases, denoting that they possessed the least artemether release retardation properties. GM0.5M1 and GM0.5C1 did, however, not adhere to the acceptance criteria for the percentage of artemether released [25], as these formulations only released 63.49% and 36.71% artemether, respectively.

No specific overall trend in artemether release profiles for formulations comprising glycerol monostearate could be defined. Nonetheless, GM0.5C1, GM0.5M1 and GM0.75C1 all depicted non-Fickian diffusion, fitting the Korsmeyer–Peppas with T_lag_ model (Table 4). The process of non-Fickian diffusion is mainly observed when the temperature is below that of glass transitioning, meaning that the polymer chains are not sufficiently mobile to permit immediate solvent penetration into the polymer core. This may be as a result of the solid lipid dispersions being prepared below their glass transition temperatures, as explained in previous work by the authors [10]. GM0.75M1 and GM1M1 followed the Peppas–Sahlin2 with T_lag_ model, whilst GM1C1 fitted the Peppas–Sahlin1 with T_lag_ model, also signifying non-Fickian diffusion (Table 4).

Cetyl alcohol formulations with MicroceLac^®^100 as filler best fitted the Korsmeyer–Peppas with T_lag_ model. CA1M1 portrayed Fickian diffusion with prominent polymer relaxation (Figure 3b). Additionally, CA1M1 depicted burst release (approximately 50%) once lag time had elapsed (Figure 2c). CA0.5M1 showed super case II transport and CA0.75M1 was classified as non-Fickian diffusion, as per the n-values. It is therefore clear that the lipid:drug ratio determined the mechanism of drug release associated with the Korsmeyer–Peppas with T_lag_ model. Cetyl alcohol-CombiLac^®^ formulations (CA1C1 and CA0.75C1) exhibited artemether dissolution in accordance with the Peppas–Sahlin2 with T_lag_ model, whilst CA0.5C1 fitted the Peppas–Sahlin1 with T_lag_ model.

Cetyl alcohol comprising formulations presented the highest average mean dissolution time values of the three lipid bases, indicating that this lipid base retarded drug release the most (Figure 2c). The effect of the presence of biorelevant media is perhaps most pronounced when considering formulation CA0.5M1, where it is clear that artemether release was constant after an increase in pH (to that simulating small intestine pH) prior to the addition of biorelevant media. CA0.5M1 and CA1C1 did not meet the acceptance criteria for percentage artemether release, as they both released less than 60% of the artemether content [25]. Cetyl alcohol-CombiLac^®^ formulations exhibited similar release profiles in terms of sustained release, and this similarity was confirmed by the fit factors (Appendix A). CA0.5C1 versus CA0.75C1 attained an f_1_-value of 4.584 and an f_2_-value of 82.168, signifying that no statistical differences were present, which coincides with these two formulations exhibiting the same lag time (150 min). However, significant differences existed between CA1C1 and the formulations CA0.5C1 and CA0.75C1, respectively, which can be explained by the difference in lag time, namely CA1C1 displayed initial drug release at 240 min, whereas both CA0.5C1 and CA0.75C1 showed signs of drug release of more than 20% at 180 min.

The inclusion of biorelevant media proved most valuable and their effects are significant. Their influence was most pronounced when considering cetyl alcohol formulations, as highlighted by an R^2^-value of 0.737 prior to their inclusion, and a subsequent R^2^-value of 0.969 thereafter. In other words, their inclusion rendered near-linear drug release. This effect was true for all three lipid bases.

Coartem^®^ comprises polysorbate 80 (a surfactant), as well as hypromellose and microcrystalline cellulose, as fillers. The presence of a surfactant, together with the gel matrix forming properties of Hypromellose, made for interesting comparisons pertaining to the percentage artemether released and the dissolution release kinetics. It was notably clear that polysorbate 80 was able to provide sufficient wetting of artemether at a lower pH of 1.2, allowing for its earlier detection comparably. Moreover, an increase in pH to 6.8 coincided with a prominent spike in artemether release from this product. Coartem^®^ fitted the Peppas–Sahlin2 with T_lag_ model, which is explained by the erosion of the tablet (visible within the first 2 min of dissolution testing) and diffusion of artemether from the gel matrix formed by the Hypromellose, as seen in Figure 3a.

During the dissolution experiments performed, lumefantrine could not be quantified for either the formulated lipid matrix tablets or the commercial product, Coartem^®^. Lumefantrine has a distribution coefficient (log D value) of 8.9 and 10.1 at pH 6 and 7.4, respectively [45]. This value reflects the distribution of a drug between an organic phase and a water phase. The considerably high log D values of lumefantrine define its affinity for an organic or lipophilic phase and provide a fundamental explanation for the absence of its detection during the dissolution studies conducted. Additionally, it was clear that the concentration and type of phospholipid included in the sequential dissolution media was insufficient to increase lumefantrine wettability to such an extent so as to produce a high enough concentration present in the water phase for detection.

To summarize, stearic acid comprising formulations provided the most consistent modified artemether release profiles, with a predictable lag time of 180 min. The inherent surfactant properties of stearic acid proved beneficial and resulted in the consistently highest initial percentage of artemether released, in conjunction with an increase in pH from 1.2 to 6.8. Thus, this lipid base was selected as optimal for the formulation of lipid matrix tablets comprising artemether. On the other hand, glycerol monostearate formulations depicted greater variances in lag times and illustrated inadequate artemether release profiles comparatively. Of the two fillers incorporated, CombiLac^®^ yielded more formulations capable of releasing the required artemether content compared to MicroceLac^®^100. Additionally, these formulations exhibited less variance in lag times. With regard to lipid:drug ratio, the 0.75:1 ratio faired favorably, as it delivered acceptable artemether amounts released for all formulations bar one (70.36%). However, in totality, SA0.5C1 bore an ideal percentage of artemether released (97.21%), according to assay value versus theoretical content. This formulation furthermore displayed the most optimal pharmacotechnical properties, taking the formulation factors and corresponding responses into account, and it may therefore be resolved that it is the most promising formulation of the investigated candidates pertaining to the pharmacotechnical properties, as well as to artemether release kinetics. In addition, when utilizing highly lipophilic active pharmaceutical ingredients, biorelevant media is of immense value and a worthwhile inclusion necessary to more accurately predict lipophilic drug release, given the more realistic physiologically mimicked environment provided.

As stated, this study focused on investigating the effect of hot fusion on augmenting dissolution, together with the significance of biorelevant media for these tested active pharmaceutical ingredients. It in no way set out to perfect a dissolution media environment optimal for the concurrent evaluation of artemether and lumefantrine. This study rather aimed to establish whether the broad, proposed simulated gastric media was specific enough to handle the unique needs of artemether and lumefantrine quantification. This study found that whilst simulated gastric media is recommended, specific method development is still required for extremely lipophilic compounds such as lumefantrine as lumefantrine release could not be detected in the dissolution media and therefore no dissolution profiles could be drawn. The general method proposed for lipophilic compounds was only successful for the detection of artemether. A discriminatory method alone, though useful, is still less than ideal due to the lack of in vivo linkage [46]. Unfortunately, when delving into method development, situations will exist that do not have quantitative relevance to all formulation variables. This study applied a broad and simplified simulated gastric media to establish the relevance of biorelevant media and to investigate the proposed dissolution method adjustments without imposing too many restrictions.

The addition of biorelevant media was vital and considerably increased the release of artemether from the lipid matrix tablets manufactured. However, the basic composition of the biorelevant media utilized lacked characteristics capable of solubilizing lumefantrine to such an extent that it could be quantitatively detected employing HPLC for the formulations manufactured during this study, or for the commercial product, Coartem^®^. This speaks to the unique needs and high lipophilicity of lumefantrine rather than to an inadequacy of biorelevant media.

### 3.4. In Vitro Permeability Studies

Permeability studies of the identified optimal formulation (SA0.5C1) were conducted as a proof of concept, and revealed a lack of specificity for the model used. To date, and to the best of our knowledge, highly lipophilic drugs incorporated into lipid-based formulations have yet to be tested extensively on the in vitro permeability model investigated. The primary drawback of diffusion chambers, specifically Ussing chambers or Sweetana-Grass diffusion chambers, is the underestimation of drug transport, particularly for lipophilic compounds [47].

The shortcomings of this model proved problematic. Only after a number of adjustments, including the addition of bile salts and phospholipids to the apical chamber to simulate the systemic environment and minimizing the dilution factor, could the conduction of a single endpoint analysis yield detection of both artemether (3.35%) and lumefantrine (4.88%) transport in the basolateral chambers. The overall average percentage of TEER reduction at time 120 min was 7.09%, indicating that the tight junctions were not opened and the tissue integrity remained intact and viable. Further ex vivo analysis of the tissue revealed 9.88% artemether and 59.56% lumefantrine drug retention within the tissue, which additionally served as an indication that both artemether and lumefantrine were released in quantifiable concentrations from the lipid matrix tablet formulation.

These results are by no means ideal. However, they do highlight the need for the development of an in vitro model suited to the specific needs of lipid-based formulations and make for interesting future prospects capable of decreasing the use of animals in research [48], whilst simultaneously providing the researcher with a more accurate permeability model for the evaluation of lipid-based formulations.

## 4. Conclusions

This study validates the feasibility of manufacturing lipid matrix tablets from solid lipid dispersions prepared by means of hot fusion. The production method employed temperatures below the melting point and glass transitioning temperatures of the active pharmaceutical ingredients, thereby only proving a lipid coating of the drugs and a lipidic microenvironment for artemether and lumefantrine to theoretically partition into during drug dissolution. Dissolution behavior testing demonstrated that the addition of biorelevant media resulted in a spike in the artemether concentration, highlighting their relevance when testing highly lipophilic active pharmaceutical ingredients. Moreover, the percentage of artemether release from the formulated lipid matrix tablets was found to be higher compared to that of the commercially available product, Coartem^®^, under the same experimental conditions. Thus, this study has provided evidence to support the plausibility of utilizing hot fusion technology to significantly augment the solubility of the antimalarial drugs analyzed, which so often fail based purely on varied bioavailability stemming from poor aqueous solubility. This, coupled with the demonstrated capability to formulate a solid oral dosage form with modified drug release, could therefore provide malaria treatment with a much-needed answer to treatment failure and the emergence of drug resistance.

## Figures and Tables

**Figure 1 pharmaceutics-13-00922-f001:**
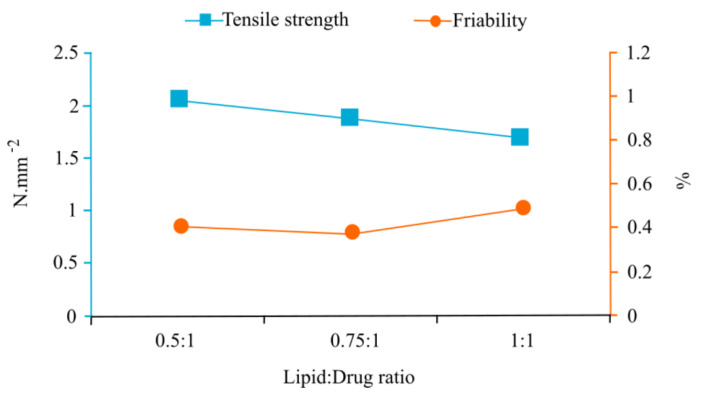
Dual-axis graph comparing lipid:drug ratio as a factor considering tensile strength (N·mm^−2^) on the left *y*-axis and friability (%) on the right *y*-axis.

**Figure 2 pharmaceutics-13-00922-f002:**
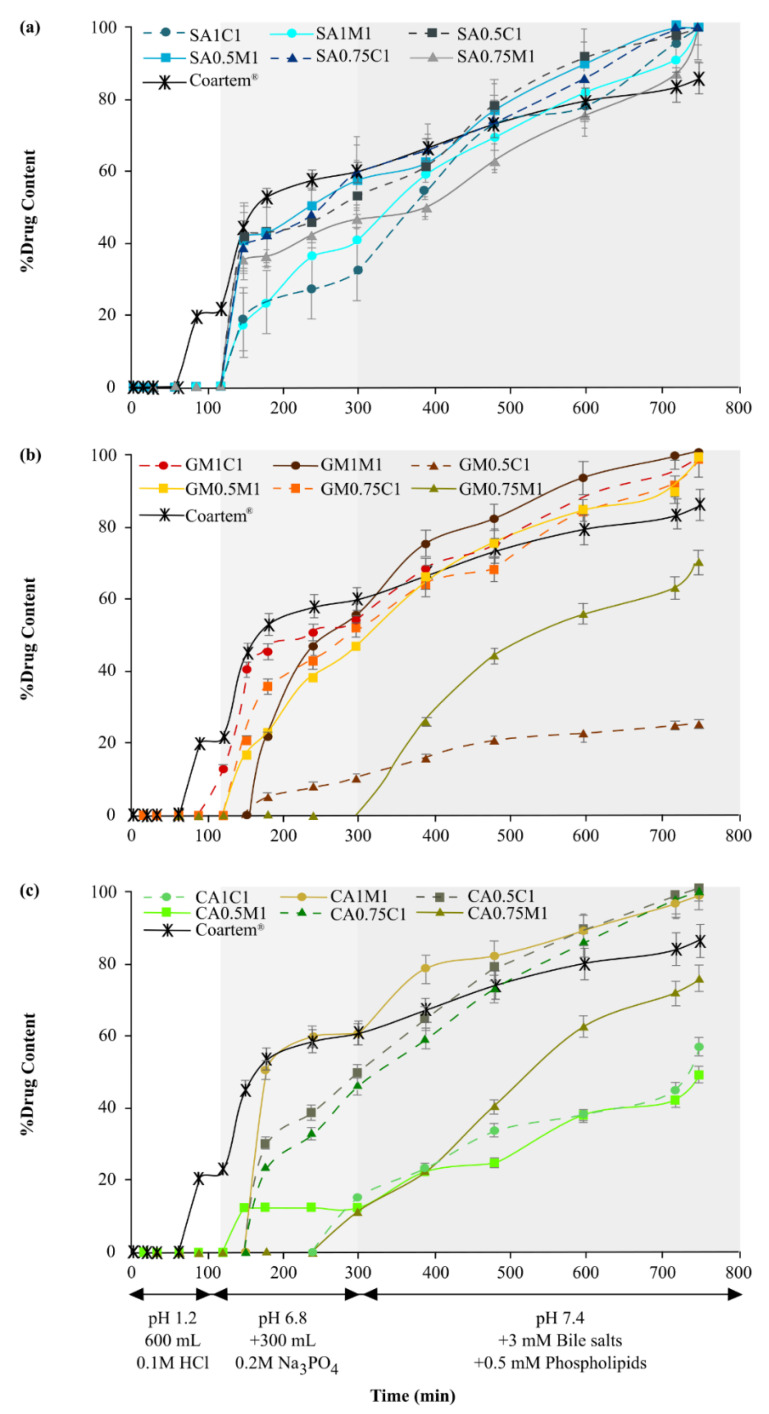
Effect of the inclusion of sequential dissolution media during dissolution behavior studies on the artemether release profiles for the respective (**a**) stearic acid, (**b**) glycerol monostearate and (**c**) cetyl alcohol formulations. The shaded areas and annotations denote the time-points for the changes in dissolution media (*n* = 6).

**Figure 3 pharmaceutics-13-00922-f003:**
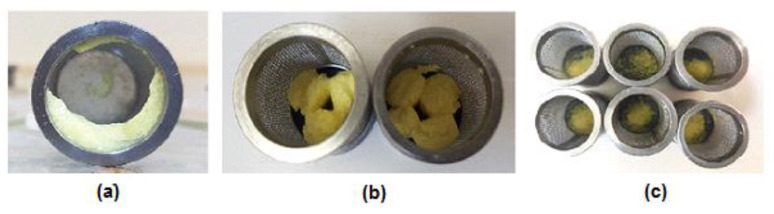
Visual representation of (**a**) hypromellose gel matrix formed by Coartem^®^; (**b**) example of relatively intact formulated lipid matrix tablets produced in this study after 750 min dissolution, highlighting the effect of polymer relaxation; and (**c**) example of formulated lipid granules that once contained active pharmaceutical ingredient particles within lipid matrix tablets, demonstrating erosion release dissolution profiles.

**Table 1 pharmaceutics-13-00922-t001:** Factorial design of independent variables X_1_ (lipid base) and X_3_ (filler), and dependent variables X_2_ (lipid:drug ratio) and X_4_ (magnesium stearate concentration), assessed during lipid matrix tablet formulation.

Exp.	X_1_	X_2_	X_3_	X_4_	Code	Exp.	X_1_	X_2_	X_3_	X_4_	Code	Exp.	X_1_	X_2_	X_3_	X_4_	Code
1	−1	−1	0	0	SA0.5C1	13	0	−1	0	0	GM0.5C1	25	1	−1	0	0	CA0.5C1
2	−1	0	0	0	SA0.75C1	14	0	0	0	0	GM0.75C1	26	1	0	0	0	CA0.75C1
3	−1	1	0	0	SA1C1	15	0	1	0	0	GM1C1	27	1	1	0	0	CA1C1
4	−1	−1	1	0	SA0.5M1	16	0	−1	1	0	GM0.5M1	28	1	−1	1	0	CA0.5M1
5	−1	0	1	0	SA0.75M1	17	0	0	1	0	GM0.75M1	29	1	0	1	0	CA0.75M1
6	−1	1	1	0	SA1M1	18	0	1	1	0	GM1M1	30	1	1	1	0	CA1M1
7	−1	−1	0	1	SA0.5C1.25	19	0	−1	0	1	GM0.5C1.25	31	1	−1	0	1	CA0.5C1.25
8	−1	0	0	1	SA0.75C1.25	20	0	0	0	1	GM0.75C1.25	32	1	0	0	1	CA0.75C1.25
9	−1	1	0	1	SA1C1.25	21	0	1	0	1	GM1C1.25	33	1	1	0	1	CA1C1.25
10	−1	−1	1	1	SA0.5M1.25	22	0	−1	1	1	GM0.5M1.25	34	1	−1	1	1	CA0.5M1.25
11	−1	0	1	1	SA0.75M1.25	23	0	0	1	1	GM0.75M1.25	35	1	0	1	1	CA0.75M1.25
12	−1	1	1	1	SA1M1.25	24	0	1	1	1	GM1M1.25	36	1	1	1	1	CA1M1.25

**Table 2 pharmaceutics-13-00922-t002:** Comparison of all formulation factors investigated in the full factorial design, and their corresponding responses per test, presented as an average for mass variation, friability and tensile strength results.

Factor	Mass Variation		Friability (%)	Crushing Strength		Tensile Strength	
	Mass (mg)	%RSD		(N)	%RSD	(N·mm^−2^)	%RSD
MicroceLac^®^100	494.939	1.390	0.429	165.41	19.611	1.850	0.148
CombiLac^®^	494.533	1.710	0.519	170.56	20.400	1.978	0.216
Stearic Acid	499.783	1.027	0.217	185.13	24.490	2.031	0.249
Glycerol Monostearate	498.592	1.912	0.209	149.58	25.886	1.613	0.086
Cetyl Alcohol	485.833	1.719	0.997	169.23	12.528	2.097	0.211
0.5:1	495.775	2.088	0.459	182.25	11.078	2.084	0.142
0.75:1	496.500	1.411	0.430	165.18	27.746	1.923	0.169
1:1	491.930	1.158	0.534	156.51	21.192	1.735	0.235
MgSt 1%	497.130	1.411	0.460	167.99	20.497	1.914	0.216
MgSt 1.25%	492.339	1.694	0.489	167.97	19.514	1.914	0.148

**Table 3 pharmaceutics-13-00922-t003:** Percentage of active pharmaceutical ingredient (API) content, presented as an average per factor evaluated.

Factor	Variable	API Content (%)	
		Artemether	Lumefantrine
Lipid type	Stearic acid	85.56	111.94
	Glycerol monostearate	83.62	92.52
	Cetyl alcohol	87.26	96.67
Filler type	MicroceLac^®^100	90.42	104.34
	CombiLac^®^	80.55	96.41
Lipid:drug ratio	0.5:1	97.58	107.20
	0.75:1	79.72	106.22
	1:1	79.14	87.71

**Table 4 pharmaceutics-13-00922-t004:** Release kinetics of artemether from lipid matrix tablets in sequential dissolution media, including biorelevant components, according to different mathematical models and fit factor values versus the release kinetics of Coartem^®^.

Formulation	MDT	Fit Factors		Goodness of Fit			Best-Fit Values	
		f_1_	f_2_	Rsqr adj	AIC	MSC	T_lag_	Model Variable
SA 0.5M1 ^a^	310.500	30.674	43.502	0.9899	97.517	3.803	120.000	k_KP_: 8.98; n: 0.37
SA0.5C1 ^a^	315.000	20.247	50.338	0.9847	104.317	3.402	120.000	k_KP_: 8.114; n: 0.385
SA 0.75M1 ^b^	382.400	39.821	38.111	0.9712	111.122	2.743	120.000	k_1_: 4.398; k_2_: 2.216; m: 0.261
SA 0.75C1 ^b^	315.500	19.180	49.743	0.9927	92.358	4.091	120.000	k_1_: 8.131; k_2_: 2.486; m: 0.244
SA 1M1 ^c^	374.000	29.238	42.157	0.9978	69.193	5.464	120.000	k_1_: 2.704; k_2_: 0.047
SA 1C1 ^c^	386.500	43.479	35.107	0.9900	94.958	3.986	120.000	k_1_: 1.956; k_2_: 0.08
GM0.5M1 ^a^	355.800	50.274	32.880	0.9968	76.064	5.094	130.442	k_KP_: 3.264; n: 0.528
GM0.5C1 ^a^	323.100	80.552	22.248	0.9951	39.170	4.685	152.871	k_KP_: 0.677; n: 0.573
GM0.75M1 ^c^	499.400	66.135	25.610	0.9981	54.831	5.756	349.090	k_1_: 4.403; k_2_: −0.051
GM0.75C1 ^a^	350.800	28.360	43.340	0.9956	81.279	4.718	120.000	k_KP_: 4.305; n: 0.481
GM1M1 ^c^	313.000	45.026	32.888	0.9990	59.719	6.239	165.510	k_1_: 5.997; k_2_: −0.075
GM1C1 ^b^	310.900	42.262	34.766	0.9920	93.219	4.026	117.890	k_1_: 10.179; k_2_: 0.823; m: 0.285
CA 0.5M1 ^a^	407.200	65.595	26.848	0.9719	85.176	2.971	45.934	k_KP:_ 0.035; n: 1.098
CA0.5C1 ^b^	341.000	30.807	40.008	0.9978	72.708	5.387	150.000	k_1_: 4.199; k_2_: 2.153; m: 0.279
CA 0.75M1 ^a^	454.100	73.660	24.011	0.9957	71.334	4.935	265.246	k_KP_: 0.570; n: 0.793
CA 0.75C1 ^c^	358.000	31.522	39.295	0.9988	60.730	6.066	150.000	k_1_: 3.456; k_2_: 0.026
CA 1M1 ^a^	285.300	23.989	42.616	0.9961	83.323	4.7571	150.000	k_KP_: 18.868; n: 0.256
CA 1C1 ^c^	472.900	72.880	24.405	0.9885	75.041	3.913	240.000	k_1_: 1.469; k_2_: 0.034

^a^ Korsmeyer–Peppas with T_lag_; ^b^ Peppas–Sahlin1with T_lag_; ^c^ Peppas–Sahlin2 with T_lag._; k_1_ = diffusion constant; k_2_ = relaxation constant; m = diffusion exponent; n = drug release exponent; k_KP_ = release constant.

## Data Availability

Not applicable.

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
