# Peer review of "Investigating In Vitro and Ex Vivo Properties of Artemether/Lumefantrine Double-Fixed Dose Combination Lipid Matrix Tablets Prepared by Hot Fusion"

_pharmaceutics, 2021, doi:10.3390/pharmaceutics13070922_

Round 1

Reviewer 1 Report

The manuscript entitled “Characterizing in vitro and ex vivo properties of artemether/lumefantrine double-fixed dose combination lipid matrix tablets prepared by hot fusion” addresses an interesting and important topic to enhance the bioavailability of antimalaria fixed-dose combination. However, there are some comments required need to be considered by the authors.

  1. The title somehow misleading it referred to the ex vivo characterization of the FDC while the results of the ex vivo as the authors conclude, it has no mean, and the development of a new model is needed.
  2. Table one is not clear. I have to back and forth in the text to understand the formulation composition and the different variables.  
  3. Table 2 summarized the tablet mass variation, friability, crushing strength and tensile strength without considering the other dependent variables. For example, the weight variation of the formulations contain microceLac, and these formulations have different rations of lipids and Mgst. Also, the RSD is missed in the friability, crushing strength and tensile strength values.
  4. Same comments (comment 3) for table 3.
  5. Error bars missed in figure 2.

Author Response

Please see attached letter

Reviewer 2 Report

The major revision of the manuscript is recommended for two reasons.

1. For the successful development of formulation the characterization of the powder materials is crucial. There is a lack in the paper any information concerning the flow and compression properties of powders used for tabletting. It must be pointed out that the use of lipid carriers for drug could drastically change the flowability of the tablet mass.
2. In the characterization of drug dissolution results, there is a lack of information concerning variability. The appropriate level o SD is a condition for use of difference and similarity factors for comparison of the dissolution profiles.

General question: the Authors apply commonly used mathematical models for fitting the dissolution data and on the base of the results the general conclusions are drawn. What does it bring to experimental approach to development of the optimized formulation?  

Author Response

Please see attached letter

Round 2

Reviewer 1 Report

The authors answered all my questions and edited the manuscript accordingly. I accept the manuscript for publication.

Thank you

Reviewer 2 Report

The proposed changes and explanations are sufficient.